# Hierarchical and Multimodal Data for Daily Activity Understanding

**Ghazal Kaviani** [1]                                          gkaviani@gatech.edu

**Yavuz Yarici** [1]                                          yavuzyarici@gatech.edu

**Seulgi Kim** [1]                                          seulgi.kim@gatech.edu

**Mohit Prabhushankar** [1]                                          mohit.p@gatech.edu

**Ghassan AlRegib** [1]                                          alregib@gatech.edu [*]

**Mashhour Solh** [2]                                          mashhour@amazon.com

**Ameya Patil** [2]                                          ameyapat@amazon.com

[1] *OLIVES at the Center for Signal and Information Processing CSIP, School of Electrical and Computer Engineering, Georgia Institute of Technology, Atlanta, GA, USA* [1]
[2] *Amazon Lab126, San Francisco, CA, USA*

**Reviewed on OpenReview:** *https://openreview.net/forum?id=0E7a6RTBqM*

**Editor:** Sergio Escalera

## Abstract

**D**aily **A**ctivity **R**ecordings for **a**rtificial **i**ntelligence (`DARai`, pronounced /Dahr-ree/), is a multimodal, hierarchically annotated dataset constructed to understand human activities in real-world settings. `DARai` consists of continuous scripted and unscripted recordings of 50 participants in 10 different environments, totaling over 200 hours of data from 20 sensors including multiple camera views, depth and radar sensors, wearable inertial measurement units (IMUs), electromyography (EMG), insole pressure sensors, biomonitor sensors, and gaze tracker. To capture the complexity in human activities, `DARai` is annotated at three levels of hierarchy: (i) high-level activities (L1) that are independent tasks, (ii) lower-level actions (L2) that are patterns shared between activities, and (iii) fine-grained procedures (L3) that detail the exact execution steps for actions. The unscripted nature of `DARai` enables the collection of action counterfactuals, defined as observed alternative executions of the same activity under different conditions (e.g., lifting a heavy versus a light object). Experiments with various machine learning models showcase the value of `DARai` in uncovering important challenges in human-centered applications. Specifically, we conduct unimodal and multimodal sensor fusion experiments for recognition, temporal localization, and future action anticipation across all hierarchical annotation levels. To showcase the shortcomings of individual sensors, we conduct domain-variant experiments that are possible because of `DARai`'s multi-sensor and and its inclusion of action counterfactuals, i.e., observed alternative executions of the same activity. The code, documentation, and dataset is available at the dedicated `DARai website`.

---

[*]. Corresponding author.

**Keywords:** Multimodal Fusion, Temporal Sequence Modeling, Cross-View Domain Adaptation, Multi-sensor Integration, Hierarchical Learning, Hierarchical Activity Recognition, Time-Series Analysis, Real-World Environments, Action Anticipation, Action Segmentation

# 1 Introduction

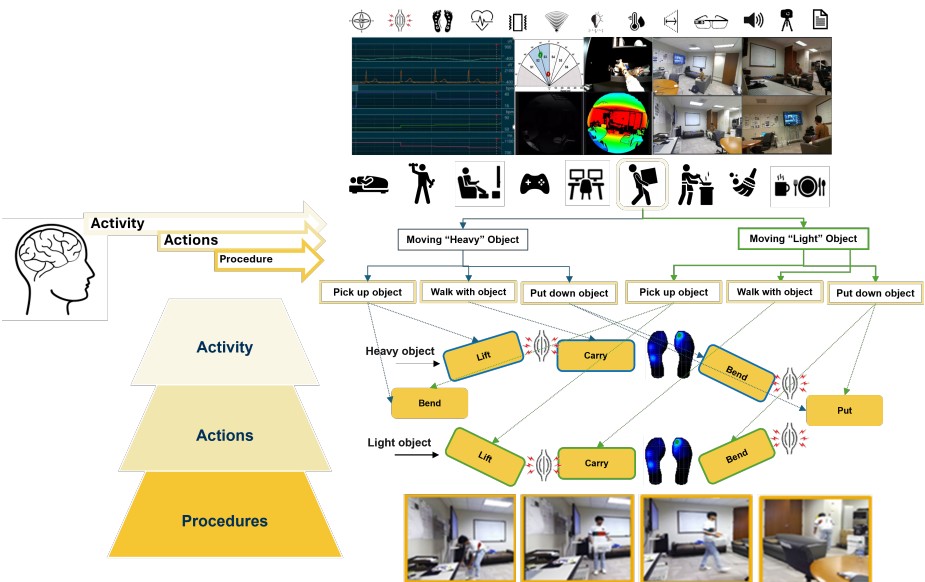

Figure 1: `DARai` multimodal data collected to capture and reflect *action counterfactual* variations across modalities. Hierarchical structure of `DARai` data, demonstrated using an example activity: Moving an Object. Two variations are shown, moving a heavy box (blue) and a light box (green), along with their corresponding lower hierarchy levels. For instance, variations in insole pressure and forearm muscle interactions are observed when lifting or carrying a box.

**Challenges in visual data understanding** Visual information has traditionally driven research in understanding human activities. Progress in computer vision techniques for tasks such as recognition and scene understanding (Kaur et al., 2024), image classification and detection (Carion et al., 2020), image segmentation (Quesada et al., 2024), video action recognition (Ma et al., 2019), and traffic surveillance (Kokilepersaud et al., 2023c) are centered around visual data. However, in real-world settings, visual processing methods face significant practical and ethical constraints, including partial occlusions (Gunasekaran and Jaiman, 2023), acquisition and environmental errors (Temel et al., 2017, 2019; Prabhushankar and AlRegib, 2022; Kokilepersaud et al., 2023d), sensitivity to angles and viewpoints (Ozturk et al., 2024; Dong et al., 2022; Garg et al., 2022), privacy concerns (Mujirishvili et al., 2023; Ravi et al., 2024), and the impracticality of maintaining fixed or unobtrusive camera perspectives. In contrast, widely available wearable sensors like inertial measurement units (IMUs), electromyography (EMG), and pressure insoles capture body-centric signals unaffected by camera placement. Furthermore, they provide additional information than visual data alone.

**Value in multimodal and hierarchical data acquisition**   Consider the activity of
Moving an object demonstrated in Figure 1. This simple activity can differ substantially
in muscle interaction, foot pressure distribution, and physiological responses depending on
whether the object is light or heavy. The differences in moving a heavy or light object may
be visually imperceptible but are clearly evident in insole pressure data as demonstrated in
Figure 1. Therefore, integrating various data modalities help mitigate the limitations of a
single modality approach (Lin et al., 2023). Such multimodal integration becomes essential
given that human activities follow a hierarchical structure. The activity hierarchy starts with
a specific goal which we term as Level 1 or L1 Activity. Moving an object in Figure 1 is an L1
activity. Next, intermediate actions, termed Level 2 or L2 are planned. These actions include
pick up, carry and put down in Figure 1. Finally, the specific procedures, including bend
and lift, that complete each action are executed. Human-centric datasets require capturing
multimodal and nested L1, L2, and L3 hierarchies that reflect both high-level and fine-grained
level sensor signals and temporal organization (Robinson and Sloutsky, 2010).

We use *action counterfactuals* to denote observed alternative executions of the same L1
activity. Two cases arise in DARai: (a) L1 activity and the L3 procedures are the same and
the L2 action sequence varies (e.g., L1 *Working on a computer* with L3 *Open a program*,
where L2 may be *take a quiz, attend a virtual meeting*, or *type an email*; L1 *Carrying an
object* with L3 *Pick up*, where L2 varies by moving a light, heavy, large, or small box),
and (b) L1 is the same and the selection or order of L2 actions and their L3 procedures
varies (e.g., L1 *Using handheld smart devices*). Actions (L2) states what steps are taken
and procedures (L3) specifies how each step is carried out; any overlap between L2 and L3
labels comes from decomposing of different complex activities into similar components, not
from similar annotations for same activity. This definition is descriptive and does not imply
causal counterfactuals.

**Dataset requirements for daily activity understanding**   Research on human-centered
applications like robotics (Chen et al., 2021), assistive technologies (Bouchabou et al., 2021),
health monitoring (Wang et al., 2023), and disease diagnosis (Prabhushankar et al., 2022)
utilize visual data and additional sparse data modalities. These additional modalities
include skeleton data (Wang and Yan, 2023), clinical and biomarker data (Kokilepersaud
et al., 2023b), and more recently language models (Damen et al., 2018) among others.
However, commonly used datasets in daily activity and action understanding emphasize
single-modality (Byrne et al., 2023) or scripted scenarios (Stein and McKenna, 2013) and rarely
incorporate hierarchical annotations across multiple sensors (Liu et al., 2019). Addressing
these gaps by designing datasets that feature continuous, unscripted recordings from both
visual and non-visual sensors can enable researchers to tackle significant challenges related to
privacy, multimodal learning strategies, and hierarchical activity recognition, thus promoting
robust, real-world human activity understanding.

In this paper, we propose the Daily Activity Recordings for artificial intelligence (DARai)
dataset. DARai is an open source, multimodal, hierarchical, and continuously recorded
dataset. DARai marks the largest available dataset interms of the number of sensors and
data modalities. The contributions of DARai are listed as follows.

1. `DARai` is a comprehensive daily activity dataset consisting of 200 hours of recordings across 20 data modalities from 12 sensors. `DARai` is collected from 50 participants in five different kitchens and living spaces, totaling 10 unique environments.

2. `DARai` is annotated to have 160 classes across three levels of hierarchies - activities (L1), actions (L2), and procedures (L3). Furthermore, an action level language description of the L1 data is provided.

3. The hierarchical structure and continuously recorded data allows construction of action counterfactual scenarios- empirically observed alternative executions of the same activity- and long-range temporal dependencies, thereby showcasing the utility of specific sensors under different conditions.

4. `DARai` is anonymized, preprocessed and curated for machine learning applications and is open sourced at IEEE Dataport (Kaviani et al., 2024).

5. Benchmarks for activity, action, and procedure recognition, action and procedure temporal segmentation, and action anticipation are provided on `DARai` and the codes are available on the project page. Across all tasks, our baseline set includes transformer-based models alongside convolutional and recurrent approaches.

**Challenges in `DARai` data collection**  Gathering continuous, real-world data across multiple environments using various sensor platforms is a significant challenge. Notably, each of the 12 sensors used in `DARai` and listed in Table 3, have different sampling rates and timestamps' protocol, making device synchronization challenging. This is addressed by implementing a multi-threaded system with a global timestamp signal to align all start and end time of data streams effectively. More information on sensor setup and calibration is provided in Appendix D. We achieve a synchronization drift of 1 ms in 24-hour timespan. Depth sensors overheat during long recording sessions, reducing depth estimation quality and requiring periodic cool downs. Wearable sensors are connected wirelessly, and sometimes encounter connectivity drops. Such connectivity issues are overcome with post-recording recovery from device internal memory, thereby necessitating using sensors that provide significant internal memory. A further challenge is the large volume of data. Each participant in a session (spanning between 30 minutes to 1 hour) generates up to 300 GB of data. This necessitates local storage before transferring processed subsets to the servers. Details on data size and format are available in Appendix C.

The novelty of `DARai` is compared against existing datasets in Section 2. The specifics of data collection, annotation, and processing are described in Section 3, followed by machine learning experiment setup and benchmarks in Sections 4, 5, 6, and 7.

## 2 Related Work

In this section, we examine 28 multimodal and hierarchical datasets to establish the novelty of `DARai` dataset.

**Multimodal Datasets**  Several widely used multimodal datasets are shown in Table 1, organized by their data collection setups: scripted activities, unscripted activities, or a hybrid

Table 1: Datasets Comparison Summary

| Setup | Dataset | # Classes | # Subjects | # Modality | Activities | Multiple Environment |
|---|---|---|---|---|---|---|
| Scripted Activities | CMU-MMAC (De la Torre et al., 2009) | 5 | 18 | 4 | Food preparation | No |
| | UT-Kinect (Xia et al., 2012) | 10 | 10 | 3 | Daily arm motions | No |
| | Multiview 3D Event (Wang et al., 2014) | 8 | 8 | 3 | Indoor actions | No |
| | UTD-MHAD (Chen et al., 2015) | 27 | 8 | 3 | Exercise, Arm motions | No |
| | Stanford ECM (Nakamura et al., 2017) | 24 | 10 | 3 | Physical activities | No |
| | Daily Intention (Wu et al., 2017) | 34 | 12 | 2 | Indoor daily motions | No |
| | UI-PRMD (Vakanski et al., 2018) | 10 | 10 | 3 | Physical rehabilitation | No |
| | NTU RGB+D 120 (Liu et al., 2019) | 120 | 106 | 4 | Indoor activities | No |
| | MMAct (Kong et al., 2019) | 37 | 20 | 3 | Indoor daily actions | Yes |
| | UESTC RGB-D (Ji et al., 2019) | 118 | 40 | 2 | Indoor aerobic exercise | No |
| | OREBA (Rouast et al., 2020) | 4 | 102 | 3 | Eating | No |
| | LaRa (Niemann et al., 2020) | 8 | 14 | 3 | Logistic and packaging | No |
| | BON (Tadesse et al., 2021) | 18 | 25 | 1 | Office activities | Yes |
| | ActionSense (DelPreto et al., 2022) | 20 | 10 | 8 | Kitchen activities | No |
| | StrokeRehab (Kaku et al., 2022) | 24 | 41 | 2 | Stroke rehabilitation | No |
| | MPHOI-72 (Qiao et al., 2022) | 13 | 5 | 2 | Indoor Human and Object Interaction | No |
| Hybrid | Breakfast Actions (Kuehne et al., 2014b) | 10 | 52 | 1 | Food preparation | No |
| | DARai (2026) | **160** | 50 | **20** | Indoor activities, Office tasks, Kitchen and Household activities | **Yes** |
| Unscripted Activities | Epic kitchen-100 (Damen et al., 2018) | 97 | 37 | 3 | Kitchen activities | No |
| | Ego4D (Grauman et al., 2022) | Varies* | 931 | 4 | Daily activities (household, outdoor, workplace, etc.) | Yes |
| | Ego-Exo4D (Grauman et al., 2023) | Varies* | 839 | 6 | Indoor skilled human activities (sports, music, dance, etc.) | Yes |
| | CAP (Byrne et al., 2023) | 512 | 780 | 1 | Indoor activities | Yes |

*Varies indicates that the exact set of activities present in the dataset is unknown and can only be determined by accessing the annotations.

approach. The data collection setup influences trade-offs between the number of activity classes, subjects, sensors setup and environments, as well as activity categories.

- Scripted Activities: Participants receive instructions to perform scripted activities. Generally, participants carry out similar activities. Such setup limits the variety in activity scenarios and environments, enabling the use of stationary and cumbersome sensors (e.g., motion capture suits).

- Unscripted Activities: Participants act freely without a scripted set of activities. Such a set-up results in unscripted or "in-the-wild" data. This setup often results in diverse set of activities and environments. Though it is difficult to maintain both consistency and variety in the sensor setup under these conditions.

- Hybrid: Participants are instructed to perform all or a subset of high-level activities from a predefined list while acting freely without a scripted plan for the lower level intermediate steps. Hybrid setups allow for more natural recordings of human activity in diverse environments while ensuring a diverse set of activities and supporting larger sensor setup.

In addition to the categorization provided in the columns of Table 1, `DARai` includes activities of varying lengths, from 30 seconds to over 5 minutes. This is opposed to existing datasets that provide short activity clips, without accounting for long-term temporal dependencies.

Table 2: Comparison of Taxonomy in Hierarchical Datasets

| Setup | Dataset | Hierarchy Structure | Hierarchy Construction Method |
|---|---|---|---|
| Post-Collection Hierarchy Design | 50 Salads (Stein and McKenna, 2013) | High-level activity, Low-level activity | Annotator decision |
| | Breakfast Actions (Kuehne et al., 2014b) | Coarse actions, Fine-grained actions | Sentences from audio transcription |
| | YouCook2 (Zhou et al., 2018b) | Activity , Steps | Sentences from subtitles |
| | FineGym (Shao et al., 2020) | Event, Set, Element | Annotator decision, Decision-tree based process |
| | MoMa (Luo et al., 2021) | Activity , Sub-Activity, Atomic Actions | Annotator decision |
| | FineSports (Xu et al., 2024) | Action, Sub-Action | Annotator decision, MixSort-OC ((Cui et al., 2023)) |
| | Ego4D Goal-Step (Song et al., 2024) | Goal, Steps, Sub-Steps | Annotator decision, wikiHow |
| Predefined Hierarchy Design | Assembly 101 (Sener et al., 2022) | Coarse actions, Fine-grained actions | Verb and noun from activity labels |
| | DARai (2024) | Activity, Action, Procedure | Predefine decomposition rules |

**Hierarchical Datasets** Unlike object detection and image classification datasets, which benefit from standard taxonomies like WordNet (Miller, 1995), human activity datasets often lack a widely adopted hierarchical or semantic framework. Existing datasets group activities under broad scenarios (e.g., daily tasks, exercise, kitchen activities) without defining multi-level hierarchies beyond activity categories. Table 2 surveys several datasets that offer explicit hierarchies and categorizes them by hierarchy setup. Post-collection hierarchy design implies that hierarchies are constructed at the annotation level after data is collected. In contrast, predefined hierarchy design allows data collection with hierarchies, inspired by activity planning and reasoning by humans. Predefined hierarchies allow variations in certain activities through multiple instances of the same activity with minor changes. For example, watching different TV programs, moving objects of varying sizes and weights, using a specific recipe to prepare food, or preparing food without a recipe are all examples of *action counterfactuals*. `DARai` consists of these action counterfactuals performed by the same participant. Predefined hierarchical structure supports more detailed analyses of complex tasks by breaking them down into smaller actions and procedures (Sloutsky, 2010).

## 3 Dataset Description and Annotation

### 3.1 Data collection process and considerations

Fifty participants were recruited through a volunteer call and recorded in ten indoor environments across five locations (living spaces, home offices, kitchens) with variation in

Table 3: Sensors and Modalities Information

| Category | Sensor | Data Modalities | Views |
|---|---|---|---|
| 3rd POV [1] | Kinect Cameraazu (2023) | RGB, Depth, IR | 2 |
| | Lidar CameraCorporation (2021) | RGB, Depth, Depth Confidence, IR | 2/1 |
| 1st POV | Microphone | Audio | 1 |
| | Wearable IMU | Wrist acceleration, gyro and magnetic Field | 2 |
| | Wearable EMG | Forearm Muscle Signal | 2 |
| | Eye Tracker | RGB, Audio, Head IMU and Gaze | 1 |
| | Insole Pressure | Total foot pressure, Multi zone pressure | 1 |
| | Wearable Bio monitors | Respiration rate, ECG, Heartbeat rate and intervals | 1 |
| Ambient and Remote | Stationary IMU | Surface vibration | 2 |
| | Radar | Doppler cubes | 1 |
| | Environmental Sensors | CO2, Humidity, Temperature, Light | 1 |
| BMI [2], Exhaustion Level | Self Report Form | Text | N/A |

lighting, time of day, air conditioning, and background noise. Each session includes *action counterfactual* instances where participants perform enforced variations (e.g., moving a heavy box instead of a light one), some of which are captured more clearly by specific sensors (e.g., insole pressure; see Fig. 1). All recordings were collected under an Institutional Review Board (IRB)–approved protocol with written informed consent; privacy safeguards and use policy are summarized in Sec. 3.4. Data is collected from 12 sensor devices, including Kinect, LiDAR, wearable IMUs on both hands, and stationary IMUs positioned on desks and floors in two environments, covering motion, physiological responses, and environmental conditions. Table 3 lists the sensors, their output modalities, and number of sensors used. Cameras are placed at different angles to capture standing and sitting postures, allowing natural movement rather than requiring participants to face a fixed camera, unlike other datasets (DelPreto et al., 2022; Liu et al., 2019; Qiao et al., 2022; Sener et al., 2022). After the recordings, participants complete two questionnaires: the first gathers information about their familiarity with the tasks, exhaustion levels, and BMI, while the second assesses their experience with the sensors. More information about the participants and the survey results are provided in the Appendix F.

### 3.2 Annotations

The annotation description of `DARai` hierarchy is provided below:

- Level 1 (L1): Activities
  High-level, independent tasks that broadly describe actions without specifying detailed goals or variations. For instance, Moving an object without defining its size or weight. Figure 2 shows the hierarchy for the L1 activity Sleeping.

- Level 2 (L2): Actions
  Actions are shorter segments that can frequently occur across multiple activities. A complete instance of an activity consists of a sequence of these actions. The order of actions or inclusion of specific steps may vary. For example, the action Prep ingredients might be skipped if ingredients are already prepared. Actions alone may lack sufficient context to definitively identify the activity. For instance, the L2 action Prepare for

Activity in Figure 2 could indicate either preparing to sleep or organizing the bed after waking, depending on temporal context.

- Level 3 (L3): Procedures

  Procedures specify the detailed fine-grained context in which actions are performed, clarifying differences between individual instances of the same action. For instance, the L3 procedures for the action Prepare for Activity in Figure 2 include Lay down and Use blanket.

- Natural language descriptions:
  Natural-language description provided by human annotators, describing activities and interactions and the scene explicitly at the L2 level.

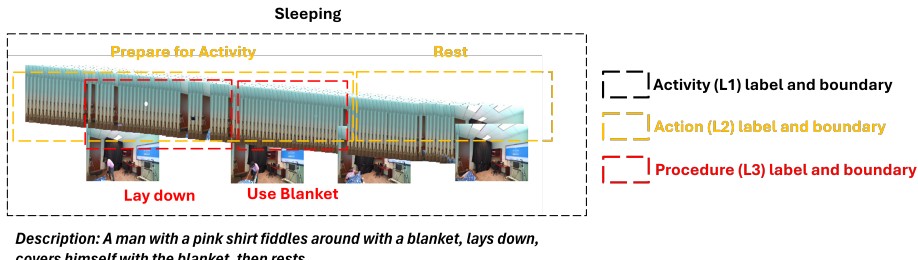

Figure 2: A sequence of frames depicting an example of an annotated activity sample: L1 activity 'Sleeping' with a sequence of L2 actions: 'Prepare for Activity' and 'Rest', and L3 procedures: 'Lay Down' and 'Use Blanket'. Boundaries indicate the start and end times of each annotation level within the corresponding upper-level video sample.

| Hierarchy Level | Number of Unique Labels | Example |
|---|---|---|
| L_1: Activities | 18 | "Moving an Object" |
| L_2: Actions | 44 | "Pick up object" , "Put down object" |
| L_3: Procedures | 98 | "Bend" , "Lift" , "Put" |

Table 4: Number of unique labels at each level of hierarchy

Some activities in the dataset do not span over all levels of the hierarchy. In Figure 2, the L2 action Rest does not contain any L3 procedures. This flexibility allows both procedural activities such as Exercise or Playing a game, and non-procedural activities including Sleeping or Watching, to fit within the same hierarchy structure and definition. In contrast, other hierarchical datasets, including (Song et al., 2024; Sener et al., 2022; Zhou et al., 2018a; Bansal et al., 2022; Kuehne et al., 2014a), have only considered procedural activities in their design. We provide the number of unique labels at each level of the hierarchy, along with examples, in Table 4.

Further details about the annotation workflow are provided in Appendix E.

### 3.3 Data Preprocessing

The `DARai` multisensory setup records over 200 hours of data across 20 modalities from 12 devices, totaling more than 20 TB of raw recordings. Appendix C and Table 11 in Appendix D.1 list formats, sampling rates, and resolutions for each sensor.

**Synchronization and alignment.** All streams are time-aligned using a global timestamp from the acquisition hosts systems and saved in addition to synchronized internal clocks, which provide consistent start/elapsed/end times for each stream. This allows segmentation and resampling without relying on device-specific timestamp protocols.[3]

**Imputation and normalization.** Short missing intervals in physiological and wearable signals are imputed by linear interpolation; gaps longer than a modality-specific threshold are left as missing and marked by a binary mask released with the data. Each channel is standardized per subject with z-score scaling, $\tilde{x}_{s,c}(t) = \frac{x_{s,c}(t) - \mu_{s,c}}{\sigma_{s,c}}$, using that subject's recordings.

Processed data are segmented into activity-level samples using L1 boundaries, with each sample corresponding to one subject performing a labeled activity within a session. Visual frames are organized with zero-padded indices; time-series signals are stored as structured CSV files with aligned timestamps across all modalities. The dataset and codes are publicly available on IEEE Dataport and Project Repository.

### 3.4 Ethics and Responsible Use

All recordings were collected under an IRB-approved protocol with written informed consent. Raw videos are anonymized frame by frame using a face detection pipeline (ORB-HD, 2023) and reviewed by annotators to flag errors; natural speech is anonymized to prevent speaker identification; Other personally identifiable information is removed except in explicitly approved demonstration cases. The dataset is released to support open-source research; users must not attempt re-identification or linkage with external sources and must follow the license terms.

## 4 Experiment Setup for ML Applications

As shown in Figure 3, `DARai` allows multimodal, hierarchical, and temporal modeling of human daily activities. In this paper, we choose three exemplar applications and demonstrate the challenges and utility of learning from human daily activity data in the following three sections. Section 5 evaluates the robustness of visual and wearable modalities under cross-view and cross-body settings, comparing fine-tuned and from-scratch deep learning models. Section 6 investigates unimodal and multimodal performance across the hierarchical levels of `DARai` for fine-grained activity recognition. Finally, Section 7 explores temporal dependencies in unscripted human activities, addressing both lower-level action localization and future activity anticipation. These tasks demonstrate `DARai`'s diverse downstream applications in machine learning.

---

3. Global host timestamps and multi-threaded capture are described in Section 1 (Challenges) and Appendix D in more details.

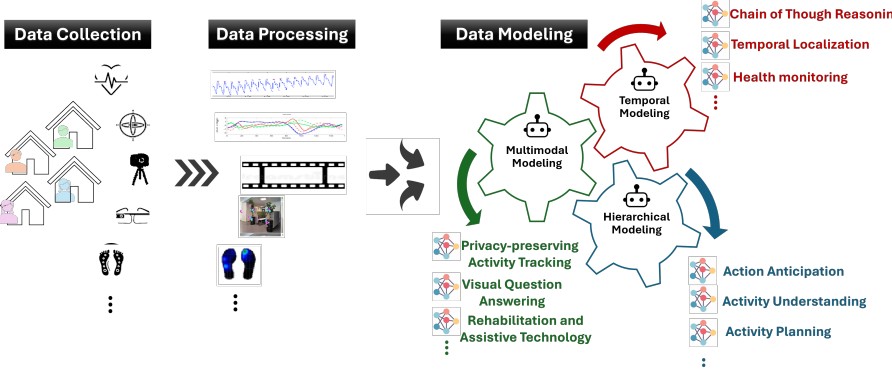

Figure 3: Overview of machine learning applications using the multimodal hierarchical `DARai` dataset. This figure highlights key aspects explored in this work, including temporal modeling, multimodal modeling, and hierarchical modeling, as well as potential applications for the machine learning community as direct or combined downstream tasks of these approaches.

**Task interfaces and benchmark templates:** To support applications beyond those evaluated in this paper, we release standardized data interfaces that work with the provided cross-subject train/test split and the temporally aligned multimodal streams. The interfaces include:

- **Time-series**: loaders and configuration templates for *supervised* and *self-supervised* tasks on IMU, EMG, insole, biomonitor, and gaze signals, that can be used individually or jointly with other modalities.

- **RGB/Depth video**: loaders and templates for video understanding tasks including but not limited to recognition, temporal segmentation, and action anticipation.

- **Video question answering**: video loaders and configuration files for open set question and answers.

The interfaces do not constrain model design or evaluation protocol. New tasks can be instantiated by adding a configuration file without modifying the dataset. The dataloaders and reference configurations are provided in the codebase.[4]

**Experiment Setup** In our experiments, we use a subset of the data modalities available in `DARai`, including RGB and Depth data from different camera views, IMU from both hands, EMG of both forearm muscles, insole pressure from feet, five biomonitoring signals, and gaze tracking data. Figure 4 summarizes the insights offered by each data modality, along with their dimensionality and resolution. The insights include motion dynamics, hand orientation, muscle activation patterns, foot pressure distribution and balance, physiological responses to varying tasks, and visual attention focus and patterns. Additional sensor data provided beyond this subset can further support research in IoT and home assistive robotics. For all experiments, we adopt a cross-subject evaluation method. Individual subjects are exclusively assigned to either the train or test set, thereby ensuring that no subject's data is used in both

---

4. https://github.com/olivesgatech/DARai

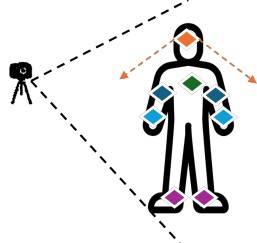

| Modality | Dimension | Resolution | Information Provided |
|---|---|---|---|
| RGB, D | 3 Channel, 1 Channel | 15 fps, 30 fps | Visual appearance, depth perception, spatial information |
| IMU | 9 Channel | 12 Hz, 200 Hz | Tracking motion dynamics and hand interactions |
| EMG | 1 Channel | 2000 Hz, 4000 Hz | Hand muscle activity , griping and holding |
| Insole | 8 Channel | 500 Hz | Insole pressure distribution and weight balancing, moving patterns |
| Bio | 5 Channel | 1000 Hz | Physiological signals |
| Gaze | 2 Channel | 200 Hz | Visual focus tracking |

Figure 4: Illustration of the unique information provided by various data modalities, highlighting the distinct insights each contributes to modeling human activity and state patterns.

training and testing. To maintain consistency across all experiments, the sets of subject IDs used for training and testing remain fixed. For reproducibility and further customization, our specific train/test subject split configuration is provided as a Python script in our GitHub repository [5]. For all classification tasks, we use class-wise top-1 accuracy. The class-wise accuracy and weighted F1 metric is chosen because the hierarchical structure of human activities does not necessarily include all actions or procedures in every instance, resulting in a nonuniform distribution of lower-level classes. More details about the class distribution can be found in Appendix E.

We highlight the key observations from our data collection, sensor setup and each experiment in Table 9. A more comprehensive version of this table is provided in Appendix B.

| Topic | Key Findings |
|---|---|
| Visual Model Training Strategies | Finetuned *visual* models *consistently outperform* those trained from scratch, across all class categories and all three levels of hierarchy.
Within kitchen and living room subsets, *depth* models trained from scratch *outperform* their RGB counterparts.
At levels L2 and L3 of the hierarchy, *depth* models continue to outperform *RGB* models trained from scratch, likely due to the stability of depth-based spatial representations across hierarchy levels. |
| Environment Differences | Activity recognition accuracy within the living room is *consistently higher* than those within the kitchen. |
| Viewpoint and Placement Effect | Visual modalities suffer a *severe drop* in accuracy when tested on the cross view.
Non-visual modalities are *robust* when the sensors capture symmetric interaction such as insole.
Wearable cross-body models show *less performance degradation* when compared to cross-view camera models. |
| Unimodal Sensor Strengths | Gaze data are effective at *separating fine-grained activities* when other data modalities that rely on physical movements suffer from performance degradation.
Bio-monitoring data are good at *identifying changes in physiology*, such as increased heart rate in Playing video games or reduced activity in Sleeping. However, they lack the resolution to separate activities with similar effort, such as different cooking activities. |
| Hierarchical Granularity | Changes in performance across granularity levels are *not* linear.
The visual model with the highest accuracy, $\sim 56\%, \sim 59\%$, experiences a *significant performance drop* when transitioning from L1 to L2 and L3.
Insole pressure and Hand EMG data modalities do *not* exhibit a significant decline in performance from L1 to L2 as visual data does.
All modality combinations experience a decline in accuracy from L1 to L3, but the magnitude of degradation *varies significantly* by sensor modality combination.
Gaze data improves L3 (procedure) activity recognition fused with other modalities. |

---

5. https://github.com/olivesgatech/DARai

| | Biomonitoring signals do not offer fine-grained insights at Level 3, even when combined with insole, EMG, and IMU data modalities. |
|---|---|
| Temporal Localization | In temporal activity localization, both the camera view and the length of the input video play a crucial role in accurately segmenting an untrimmed activity sample into lower-level segments. Actions and procedures are not isolated events. They are interconnected components of larger sequences, with a strong dependence on the temporal context of those that come before and after. |
| Short- and Long-Horizon Anticipation | At L2 (Action level), a moderate observation period provides sufficient temporal context for immediate action anticipation, while excessive observation may introduce redundant or less relevant information. Long-horizon anticipation at the L2 level, however, remains consistently lower and exhibits minimal improvement as the observation rate increases. This indicates that action-level predictions do not benefit significantly from extended temporal context, possibly due to the more independent nature of individual actions. At the L3 (Procedure level), short-horizon anticipation improves as the observation rate increases, reaching optimal performance around a 70% observation rate. Long-horizon anticipation at this level consistently improves with increasing observation rates, eventually surpassing short-horizon performance beyond a 70% observation rate. This trend suggests that fine-grained procedural activities exhibit stronger temporal dependencies, benefiting from longer observation windows that provide richer contextual information. |

Table 5: Key observations

## 5 Visual Data Robustness and Real-World Limitations

Visual sensing modalities, such as RGB and depth, are effective for human activity recognition (Ma et al., 2019). However, such modalities suffer from robustness challenges in uncontrolled settings (Hara et al., 2021; Ponbagavathi et al., 2024). In particular, existing benchmarks that rely on vision modalities are generally limited to coarse-grained actions and do not address real-life scenarios. In contrast, `DARai`, with diverse modalities and multiple levels of hierarchical labels, was designed and curated to provide a benchmark for real-life deployment. This section presents a benchmark for evaluating the robustness of visual activity recognition models across varying viewpoints and environments.

### 5.1 Visual Data Benchmarking

Given the rise of large-scale pretrained models, we examined whether pretraining compensates for the diversity of daily activities. We compared models trained from scratch on `DARai` with models first pretrained on Kinetics 400 and then fine-tuned on `DARai`. For this benchmark, we evaluated R3D (Tran et al., 2018), MViT-small (Li et al., 2022), and Swin-tiny (Liu et al., 2022). Table 19 in Appendix G compares their top-1 accuracy on combined and individual environment classes, while Table 6 reports performance by camera view at hierarchy levels L2 and L3. Although accuracy is high at the activity level (L1), all models show a notable drop at finer-grained levels (L2, L3).

Analyzing these results presented in Table 19 and Table 6 reveals a number of key observations:

**Finetuned vs. Trained from scratch**  Pretrained models consistently outperform those trained from scratch at all hierarchy levels. This advantage is most evident when training data are limited, where fine-tuned models leverage diverse prior features for higher accuracy.

**Environment-Specific Activities**  Recognition accuracy in the living room is consistently higher than in the kitchen. Even after fine-tuning on kitchen data, pretrained models still

| Data | Class Category Camera View | Level | Model | All Accuracy - Finetuned | Accuracy - Trained from scratch | F1 - Finetuned | F1 - Trained from scratch |
|---|---|---|---|---|---|---|---|
| RGB | 1 | L2 | ResNet | $0.46 \pm 0.006$ | $0.31 \pm 0.01$ | 0.458 | 0.295 |
| | 2 | | | $0.39 \pm 0.01$ | $0.25 \pm 0.02$ | 0.375 | 0.266 |
| | 1 | | Mvit-s | $0.43 \pm 0.032$ | $0.07 \pm 0.004$ | 0.418 | 0.303 |
| | 2 | | | $0.38 \pm 0.016$ | $0.14 \pm 0.07$ | 0.382 | 0.12 |
| | 1 | | Swin-t | $0.42 \pm 0.006$ | $0.18 \pm 0.02$ | 0.411 | 0.143 |
| | 2 | | | $0.33 \pm 0.03$ | $0.10 \pm 0.021$ | 0.325 | 0.064 |
| | 1 | L3 | ResNet | $0.52 \pm 0.006$ | $0.35 \pm 0.01$ | 0.489 | 0.318 |
| | 2 | | | $0.45 \pm 0.007$ | $0.28 \pm 0.01$ | 0.431 | 0.255 |
| | 1 | | Mvit-s | $0.53 \pm 0.01$ | $0.14 \pm 0.07$ | 0.507 | 0.070 |
| | 2 | | | $0.44 \pm 0.02$ | $0.05 \pm 0.003$ | 0.418 | 0.073 |
| | 1 | | Swin-t | $0.47 \pm 0.02$ | $0.25 \pm 0.02$ | 0.446 | 0.213 |
| | 2 | | | $0.34 \pm 0.01$ | $0.17 \pm 0.016$ | 0.321 | 0.138 |
| Depth | 1 | L2 | ResNet | | $0.56 \pm 0.02$ | | 0.525 |
| | 2 | | | | $0.53 \pm 0.01$ | | 0.487 |
| | 1 | | Mvit-s | - | $0.55 \pm 0.02$ | - | 0.560 |
| | 2 | | | | $0.53 \pm 0.02$ | | 0.510 |
| | 1 | | Swin-t | | $0.41 \pm 0.01$ | | 0.401 |
| | 2 | | | | $0.40 \pm 0.05$ | | 0.404 |
| | 1 | L3 | ResNet | | $0.42 \pm 0.01$ | | 0.395 |
| | 2 | | | | $0.38 \pm 0.01$ | | 0.344 |
| | 1 | | Mvit-s | - | $0.33 \pm 0.11$ | - | 0.290 |
| | 2 | | | | $0.29 \pm 0.16$ | | 0.259 |
| | 1 | | Swin-t | | $0.25 \pm 0.12$ | | 0.211 |
| | 2 | | | | $0.24 \pm 0.02$ | | 0.215 |

Table 6: Top-1 accuracy results for visual data experiments using three model architectures on Action and Procedure level of DARai hierarchy. Since activities across subsets may share similar actions or procedures, they cannot be separated by activity class subset. Results are reported for all available actions and procedures classes. Fine-tuned experiments are performed using models pretrained on Kinetics 400 dataset.

maintain a notable performance gap, suggesting stronger alignment with living-room-style activities.

**Fine-Grained Levels (L2 and L3)** At L2 and L3, trained from scratch depth models surpasses RGB counterparts, likely due to stable spatial representations. However, once using pretrained weights, RGB models achieve higher performance.

## 5.2 Robustness to Real-World Conditions

### 5.2.1 VISUAL CROSS-VIEW EVALUATION

`DARai` provides real-world conditions to test model robustness. For example, Figure 5 shows cross-view evaluations on three models, namely ResNet, MViT-s, and Swin-t, trained on a camera view and tested on same-view versus cross-view using RGB and depth data, respectively. As expected, these results show that models suffer a severe drop in accuracy when tested on the cross view. Objects and subjects that are fully within the viewpoint of a camera may not necessarily be viewable by the other camera. Thus, a significant drop in performance, e.g., ResNet performance drops from 89% to 15% and same model trained on the second view accuracy drops from 72% to 5%, is expected. Interestingly, depth data shows a similar trend although depth provides view-invariant information such as distances. This can be explained by the occlusion across views. Overall, vision-based recognition is tightly coupled to the camera perspective, a serious drawback for daily life applications within a home environment.

### 5.2.2 WEARABLE CROSS-BODY EVALUATION

While cross-view generalization is a common concern for RGB and depth modalities, wearable sensors face both cross-subject and cross-body challenges (Yarici et al., 2025). In this analysis,

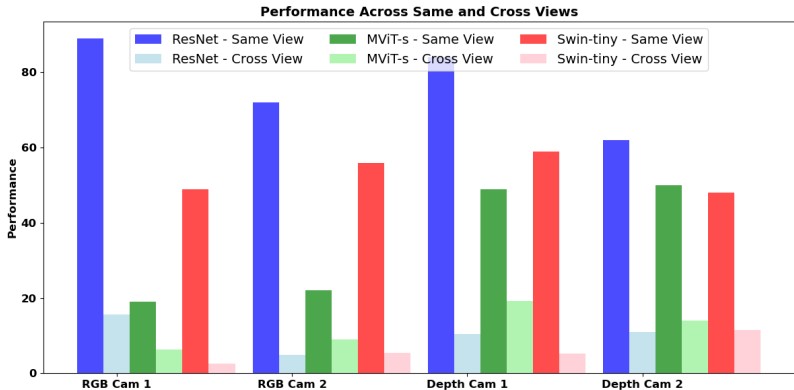

Figure 5: Comparison of same-view and cross-view inference performance for ResNet, MViT-s, and Swin-t using visual data. The results show top-1 accuracy for each model when tested on the same camera view and cross views.

data from a sensor worn on one side of the body, e.g., a hand, an arm, or a foot, is used for training while testing is performed on data from the same type of sensor on the other side of the body. This setup creates a "cross-body" domain shift. In such settings, the challenges are different from those in vision-based cross-view systems. Certain body activities are mostly symmetric across the left and right parts of the body. Those activities are generally related to movement patterns and balancing body weights and pose, represented by foot-pressure insoles as depicted in Figure 6 for the Insole sensors. In contrast, hands interactions are generally asymmetric and one hand is usually dominant. As a result, a model trained on right-wrist or right-hand signals may struggle when tested on data from the left side, which can have a weaker or distinct signal, leading to higher error rates. This is clearly depicted in Figure 6 for the IMU and EMG sensors.

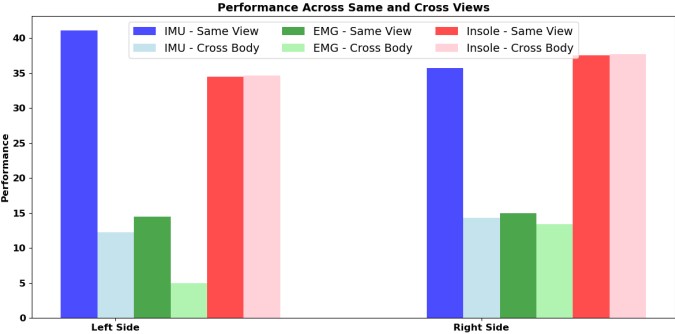

Figure 6: Comparison of same-view and cross-view inference performance using IMU, forearm muscle EMG, and insole pressure wearable data. The results show top-1 accuracy for each sensor when tested on the same side of the body and the opposite side. Models were trained separately on left-side and right-side data for this experiment.

In general, based on DARai, wearable cross-body models show less performance degradation when compared to cross-view camera models. This observation leads to the question about

the contribution of every modality to recognition at every hierarchy. There seems to be a collaboration across these modalities to create a representation space that outperforms the counterparts created by a single or a subset of these modalities. `DARai` provides the proper settings to conduct such investigation.

## 6 Multimodal and Hierarchical Human Activities Benchmark

In this section, we evaluate unimodal and multimodal approaches on `DARai` across the three hierarchical levels (L1, L2, L3). We employ a transformer-based architecture, introduced in Appendix G, to analyze each modality alone and in pairwise combinations. Figure 7 presents these experiments results, highlighting both individual performance differences and the advantages of fusing complementary data modalities.

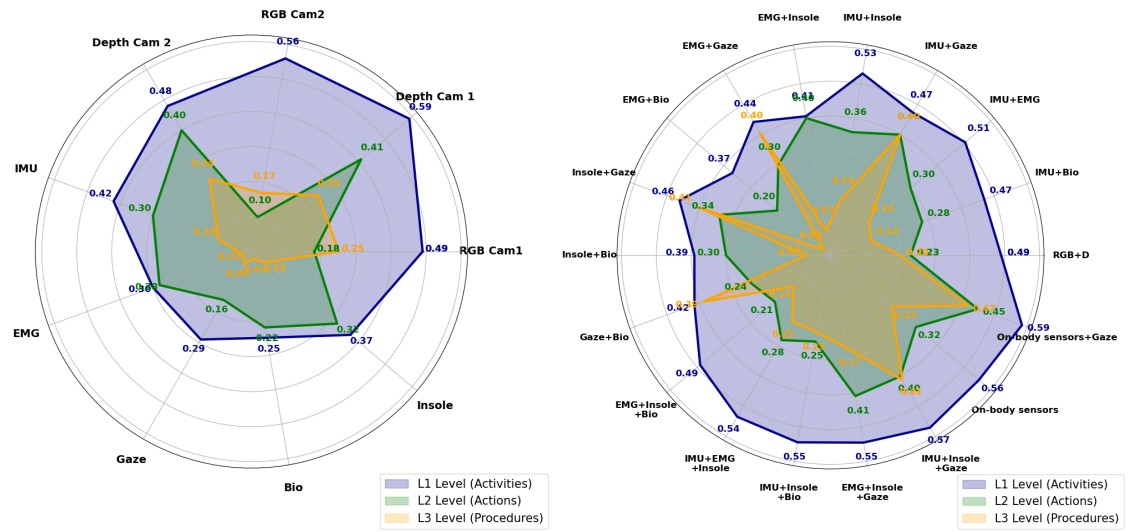

(a) A transformer-based architecture was trained from scratch on each modality separately for this experiment, using Swin-t for visual data modalities and the modified transformer introduced in Appendix G for other data modalities.

(b) A transformer-based architecture was trained to fuse each modality pairs separately for this experiment, using Swin-t for visual data modalities and the modified transformer introduced in Appendix G for other data modalities.

Figure 7: Comparison of Top-1 accuracy results for individual data modalities and data modality pairs across 3 level of `DARai` hierarchy.

## 6.1 Unimodal Analysis and Comparison

Table 7 summarizes the effectiveness of different sensing modalities in recognizing human activities. Each modality excels in specific contexts but also has limitations. For example, gaze data is effective for activities involving sustained visual fixation, while EMG signals capture distinct muscle activation patterns. Biomonitoring data provide insights into physiological changes but lack fine-grained resolution for differentiating similar exertion levels. Given such complementary effectiveness, it is common to fuse these modalities to improves accuracy and robustness. We show the confusing matrices of uni-modal data and fusion of five uni-modal

data on Figure 8. The confusion matrices show that each sensor captures unique aspects of activity but struggles when used alone. For instance, Insole data effectively captures unique foot pressure patterns and performs well in recognizing activities such as Carrying objects, where additional weight affects pressure distribution. On the other hand, relying on Insole modality only introduces uncertainty in differentiating between activities such as Cleaning dishes and Making a salad that have similar standing posture. Another example is IMU data, which are effective at separating activities with dynamic movements such as Exercising and Working on a computer, which is a primarily static activity. However, IMU data are less effective in separating activities with similar hand movements such as Cleaning dishes and Cleaning the kitchen. More detailed study on fusion based on DARai is in Appendix G.

| Modality | Best Suited For | Example Activities |
|---|---|---|
| Gaze | Sustained visual fixation | Reading, Watching TV, Working on a computer |
| EMG | Muscle activation patterns | Making pancakes (whisking motion), Playing video games (controller grasping) |
| Bio-Monitoring | Physiological variations | Playing video games (increased heart rate), Sleeping (reduced activity) |
| Insole | Planter pressure patterns | Carrying objects (added weight changes foot pressure), Standing postures |
| IMU | Dynamic body movements | Exercising (dynamic motion), Working on a computer (static) |

Table 7: of Different Modalities for Activity Recognition

DARai's unique hierarchical and multimodal aspects provide a space to investigate modality contribution to every level of the hierarchy. Figure 7a shows the performance of each of the modalities. The best-performing unimodal models for non-visual data modalities are shown on this plot. The input length, window size, and other parameters are kept consistent across all levels of hierarchy, with samples drawn exclusively from one hierarchy level at a time (L1, L2, or L3). As shown in Figure 7a, changes in performance across granularity levels are *not* linear. Furthermore, different modalities respond differently to decreasing abstraction, shorter sample sequences, and finer-grain details. Previous studies have shown that task granularity negatively impacts natural image models (Prabhushankar et al., 2020; Kokilepersaud et al., 2024, 2025). Our findings confirm that even the visual model with the highest accuracy (i.e., 56%-59%), experiences a significant performance drop when transitioning from L1 to L2 and L3. In contrast, insole pressure, forearm muscle EMG and biomonitoring modalities do not exhibit the same steep decline in performance, experiencing only 5%, 2%, and 3% drops from their L1 accuracy, respectively. Notably, the decline in visual model performance is smaller between L2 and L3, with both camera views performing better at L3 than at L2. This trend is expected, as 20% of the procedures (L3) are shared between higher-level activities (L1) and actions (L2), indicating more overlap at these levels.

In contrast, biomonitor signals and hand muscle EMG signals demonstrate greater resilience to increased granularity between L1 and L2. Additionally, insole pressure outperforms other non-visual modalities at the action level (L2) with 32% accuracy. The alignment between plantar patterns and the structure of actions makes insole data effective for recognizing fine-grain actions at this level. However, from L2 to L3, none of these modalities maintain the same level of resilience. In naturalistic daily life activities, data are not trimmed around a single action, so wearable time-series often include extended inactive intervals, such as minimal hand movement or prolonged standing in one signal, while other modality signals remain more active. These intervals can overshadow distinct activity patterns within a single

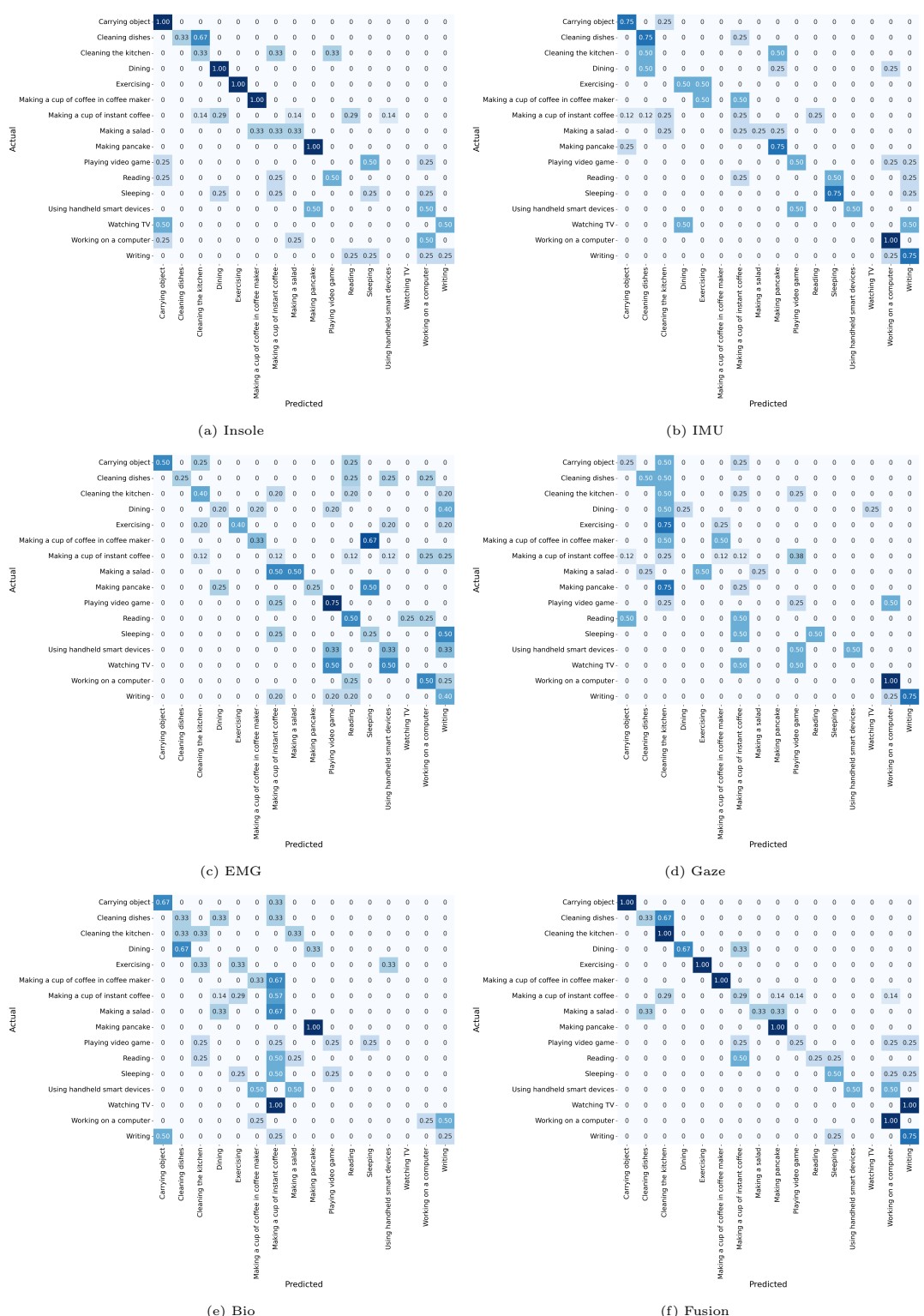

Figure 8: Activity confusion matrices for unimodal and multimodal settings.

modality and lead to performance decline. At lower levels of the hierarchy, the increased granularity magnifies this effect, making it harder to capture pattern differences.

### 6.2 Multimodal Analysis and Comparison

We evaluate multimodal modeling across the three hierarchical levels of `DARai`, considering both pairwise combinations (Figure 7b) and structured sensor groups (Table 8). All modality combinations exhibit decreasing accuracy from L1 to L3, but the extent of the decline varies by modality type. Pairwise results (Figure 7b) show that some modalities complement each other effectively, such as IMU with Insole or Gaze, while others provide limited benefit when fused with others, such as EMG with biomonitoring. These examples illustrate how the information carried by different signals overlaps or reinforces each other, but the overall trends are clearer when modalities are considered in broader groups.

Table 8 summarizes performance when sensors are ablated into functional groups. Physiological signals (EMG and biomonitoring) are informative at coarse granularity (L1) but their performance declines sharply at finer levels, indicating limited discriminative power for detailed activities. Biomechanical signals (IMU and Insole) achieve stronger results and degrade more gradually, reflecting their utility in capturing motion dynamics. Gaze contributes to attentional cues but performs modestly when used alone. Third-person vision (RGB and Depth) provides competitive performance at L1 but deteriorates substantially at finer levels, highlighting the difficulty of translating scene context into fine-grain distinctions. The best overall performance is achieved when combining all wearable sensors (physiological, biomechanical and behavioral), which sustains relatively high accuracy across all levels, reaching over 42% at L3.

These findings suggest that while individual modalities have characteristic strengths and weaknesses, grouping them by functional role highlights which categories are robust to task granularity. Physiological signals alone are fragile, external vision is coarse-grain biased, and biomechanical dynamics are more stable, but fusing multiple wearable signals consistently yields the most reliable performance across the hierarchy.

| Placement | Sensor Group | Modalities | L1 Accuracy | L2 Accuracy | L3 Accuracy |
|---|---|---|---|---|---|
| On-body | Physiological | EMG + Bio Monitor | 0.368 | 0.200 | 0.026 |
| | Biomechanical | IMU + Insole Pressure | 0.530 | 0.359 | 0.159 |
| | Behavioral (attention) | Gaze | 0.298 | 0.159 | 0.052 |
| | All | EMG + Bio + IMU + Insole + Gaze | 0.585 | 0.451 | 0.423 |
| Off-body | Third-Person Visual | RGB + Depth | 0.49 | 0.23 | 0.20 |

Table 8: Comparison of Different Sensor Modality Groups. Missing entries correspond to cases where no single-modality results were available.

## 7 Temporal Dependencies Benchmark

Understanding temporal dependencies is crucial for modeling human activities, which unfold hierarchically over time. Unlike existing datasets with predefined step boundaries, DARai presents a continuous, unstructured setting where activities transition fluidly across different granularities, posing challenges in recognizing hierarchical structure and temporal

dependencies. To assess these dependencies, we focus on temporal activity localization and activity anticipation. Performance is evaluated using Mean over Classes (MoC), which measures per-class performance and averages across all classes, following the long-term action anticipation protocol (Gong et al., 2022; Abu Farha et al., 2021; Sener et al., 2020; Ke et al., 2019; Kim et al., 2026). For consistency, results are averaged over three different seeds. For the activity localization result, refer to G.2.

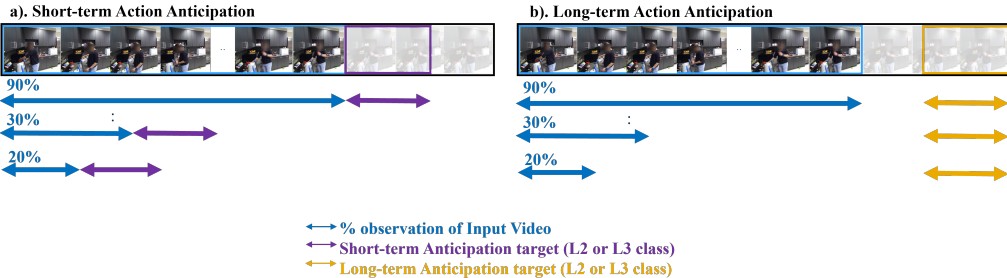

Figure 9: Feeding different length of sample sequence for action anticipation task. (a) shows short-term action anticipation task, and (b) shows long-term action anticipation task.

## 7.1 Short- and Long-Horizon Activity Anticipation

Action anticipation predicts future actions from partially observed sequences. `DARai` presents extended, fine-grained actions and procedures from a third-person view, enabling two tasks: **short-horizon**, which focuses on immediate upcoming actions, and **long-horizon**, which targets more distant actions. Figure 9 illustrates the setup, and we vary the observation ratio $\alpha$ from 0.1 to 0.9.

Figure 10 shows that short-horizon anticipation improves steadily with more observation—up to about 50%—but gains little beyond that. This indicates that a moderate window captures enough context, and extra observations can introduce redundancy. In contrast, long-horizon anticipation consistently benefits from increasing the observation ratio, reflecting the strong temporal dependencies of extended procedural activities in `DARai`.

## 7.2 Hierarchical Activity Anticipation

Finer-grain anticipation tasks add additional complexity, as the model must understand both the hierarchical structure and the sequence of events or actions from higher-level activities (Kim et al., 2025). This type of anticipation is far more challenging than traditional feature prediction based on observed features. To address such challenges, `DARai` is essential for testing and benchmarking models, since it provides multiple levels of hierarchy.

Figure 10 shows the disparity in anticipation performance across different levels of temporal granularity. At the action level (L2), short-term action anticipation accuracy fluctuates between 42% and 50%, while long-term anticipation remains lower, ranging from 20% to 25% across different observation rates. This suggests that action anticipation primarily relies on local temporal cues rather than extensive historical context. Conversely, at the procedure level (L3), long-term anticipation accuracy at L3 improves from 36% to 46%, while

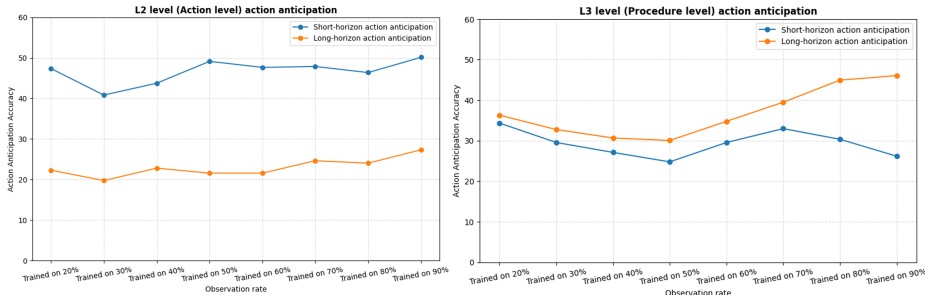

Figure 10: Action anticipation accuracy across different observation rates based on partially observed L1-level sequences at L2 (Action level) and L3 (Procedure level). The left graph represents L2-level action anticipation for both anticipating over the next 8 seconds (short-term action anticipation) and the latest 8 seconds of each sequence (long-term action anticipation). The right graph represents L3-level action anticipation for both short-term and long-term action anticipation. The accuracy represents Mean over Classes (MoC) accuracy.

short-term varies more significantly, ranging from 26% to 34% as observation rate changes. In addition, the confusion matrices of Figure 11 shows that predicting the next L3 labels (b) is harder than predicting the next L2 labels (a). Specifically, from (a) we infer that the model confuses activities within the same category, such as "Take out smartphone" vs. "Using smartphone" in device-related tasks, and "Prepare ingredients" vs. "Get ingredients" in kitchen-related tasks. These findings emphasize the importance of considering the hierarchical nature of activities when designing anticipation models, as the required observation window and predictive capability vary depending on the level of granularity. Thus, optimizing anticipation strategies should be guided by the underlying temporal structure of the target activities.

## 7.3 Activity Anticipation in Action Counterfactual cases

We further conduct experiment on action counterfactual cases. Especially, we select pairs of samples that share identical L1 and L3 labels while differing only in their L2 labels. Specifically, both cases in Figure 12 (a) share the same L1 label "Carrying Object" and L3 labels ("Pick up box", "Walk with box", "Put down box") but differ in L2 labels ("Carrying light/small object" vs. "Carrying heavy/big object"). In (b), both share same L1 label "Reading" and L3 labels ("Get paper/book", "Read paper/book") with L2 differences ("Reading at desk" vs. "Reading on couches"). In (c), both share same L1 label "Playing Video Game" and L3 labels ("Playing reaction game", "Playing speed game") with L2 variations ("Playing on TV" vs. "Playing on computer").

To examine the model's generalization ability to unseen counterfactual scenarios, during training, only the factual cases were included in the dataset, while the counterfactual cases were exclusively used during inference for comparison. The objective of the experiment is to predict the next L3 labels.

The results reveal distinct performance patterns: In cases (a) and (b), significant performance gaps were observed between factual and counterfactual samples, whereas in case (c), the model performed similarly in both scenarios. Notably, in case (b), the model frequently misclassified counterfactual samples as "Conversation on the phone" or "Rinse dishes". We

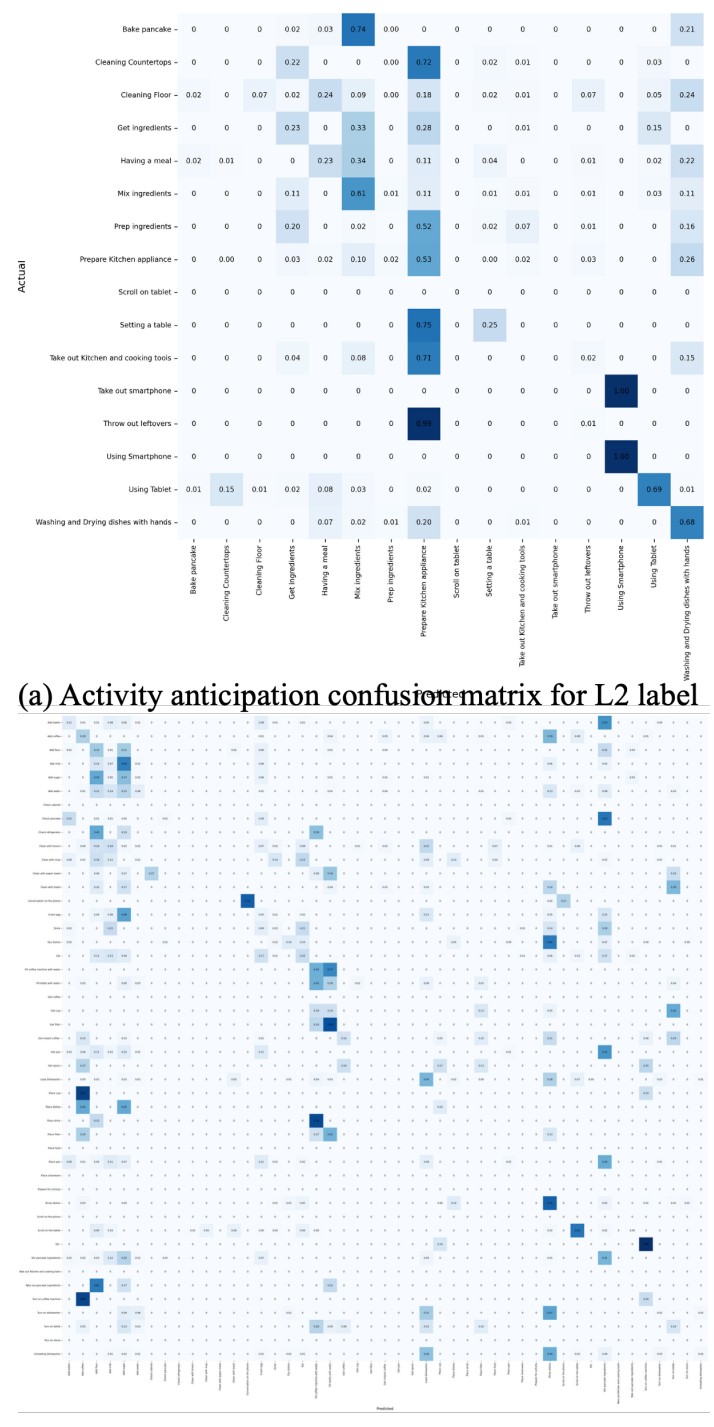

(a) Activity anticipation confusion matrix for L2 label

(b) Activity anticipation confusion matrix for L3 label

Figure 11: This figure shows the comparison of confusion matrices demonstrating action anticipation performance when predicting next L2 labels (a) and L3 labels (b).

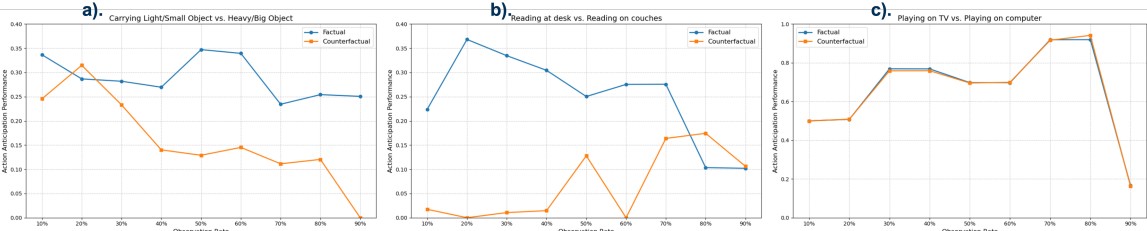

Figure 12: Action counterfactual cases demonstrating performance comparisons under different observation rates. (a) Comparison between 'Carrying light/small object' (factual) and 'Carrying heavy/big object' (counterfactual). (b) Comparison between 'Reading at desk' (factual) and 'Reading on couches' (counterfactual). (c) Comparison between 'Playing on TV' (factual) and 'Playing on computer' (counterfactual). The graphs show action anticipation performance across varying observation rates from 10% to 90%.

attribute this error to the model's inherent bias towards contextual background patterns, as "Conversation on the phone" in the dataset often occurs in couch settings. This indicates that the current state-of-the-art models, when applied to the `DARai` dataset, tend to rely heavily on contextual biases rather than focusing on the intrinsic characteristics of the target actions themselves.

## 8 Discussion

**Activity structure** Continuous, unscripted recordings include long inactive intervals and fluid boundaries between actions and procedures. This complicates segmentation and recognition but reflects natural behavior.

**Action counterfactuals** Providing activity categories without scripts leads participants to produce alternative executions of the same activity. This changes temporal relations between actions and procedures and supports analysis of execution variability.

**Granularity** Performance shifts across L1, L2, and L3 are not uniform. Coarse activities are easier to classify than fine-grained actions and procedures. Insole pressure and EMG are less sensitive to granularity than RGB.

**Robustness** Models trained on a specific camera view or body side lose accuracy when tested on a different view or side. This shows the need for methods that handle placement and viewpoint changes. Sensor fusion improves accuracy when single modalities are weak.

**Temporal modeling** Actions show higher temporal diversity than procedures. Longer observation windows help localize actions. Predicting long-horizon procedures is often easier than predicting long-horizon actions.

**Limitations** The large sensor setup reduces flexibility for data collection and expansion. As a result we based our study on a cohort of 50 participants. We acknowledge that while this scale is on par with leading scripted and hybrid datasets, it does not match the participant breadth of recent large-scale, 'in-the-wild' collections. Future work can broaden environment coverage and explore fully wireless sensor setups. While the custom portable system cost

was approximately $22,000, more economical alternatives may be employed, as the setup and results are not contingent upon specific sensor brands.

**Broader Impact**  By showing that non-visual modality fusion improves recognition, `DARai` supports privacy-focused approaches with minimal or no camera use. The dataset encourages development of models that are robust to placement and viewpoint changes. The data release follows informed consent of participants and promotes open research. At the same time, human activity datasets inherently raise privacy considerations. To address this, we exclude personal identifiers and blur faces where possible. These safeguards contribute to, but do not resolve, the broader challenge faced by the research community in balancing data utility with privacy.

## 9 Conclusions

We introduced and open-sourced `DARai`, a hierarchical, multimodal dataset with continuous, unscripted recordings and with 3 level (L1,L2,L3) temporal annotations. We reported benchmarks on robustness (cross-view and cross-body), multimodal fusion, and temporal modeling (localization and anticipation), and outlined additional tasks enabled by the data and dataset loader and configurations in the project repository.

## 10 Acknowledgments and Disclosure of Funding

This work is partially funded through the support from the Franklin Foundation via the John and Marilu McCarty Chair professorship and a gift from Lab126 at Amazon.

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
