# OpenReview forum: "Hierarchical and Multimodal Data for Daily Activity Understanding"
_DMLR — Accepted by DMLR_

### Review · Reviewer_pyco · 2025-05-08

**Recommendation:** 3
**Confidence:** 2

**Summary Of Contributions:**

Paper proposes a new dataset for daily activity recognition (DAR) studies, including multiple sensor modalities and hierarchical labelling of the activities (up to 160 different classes in different hierarchy levels), collected from 50 participants in indoor environment setups, and with scripted and unscripted activities. More specifically, activities are divided on 3-level hierarchy of high-level activities (L1), middle-level actions (L2), low-level procedures (L3). Proposed datasets is compared to other existing repositories, current DAR challenges are analysed and discussed, and benchmark/baseline ML experiments are performed on the topics of model robustness, multimodal and hierarchical human activity recognition, and temporal activity anticipations, showing the properties of the data and challenges in the DAR domain.

**Strengths:**

Paper presents interesting dataset contribution to multimodal human activity recognition benchmarking. Although there have been many previous datasets, proposed setup brings some new variations to benchmarking with large number of classes, subjects, modalities, and how the activities have been collected. Overall, it provides valuable evaluation datasets for ubiquitous computing and ML communities. Paper is, most of the parts, clearly written and structured.

**Audience:**

Yes

**Broader Impact Concerns:**

Broader impact statement is present in the paper, but could include more detailed description and analysis of privacy concern related to
the human daily activity and behaviour data.

**Claims And Evidence:**

Most of the claims are supported: 1) introducing the new hierarchical daily activity datasets with large number of classes, subject, and modalities longer time-series, collected from different indoor environments with proper web links and access details for publicly available dataset, 2) initial benchmark, showing the three different aspects of the datasets and research challenges in this domain.

As stated above, there are few weaknesses and proposed adjustments that could improve the representation, re-usage of the dataset, and
reproducibility of the benchmark results.

**Datasets And Benchmarks:**

Submission includes sufficient details on:
- Data collection
- Sensor

What could be improved:
- Data pre-processing could include more details, e.g., imputation, normalisation, segmentation
- Some of the details could be moved from appendix to main text related to dataset availability, maintenance etc.
- Ethical and responsible use could be emphasise better (how human-based data is handled)
- Dataset link in the appendix does not work; from Github repository you can access to it

**Extended Submissions:**

N/A

**Limitations:**

See requested changes above.

**Requested Changes:**

A few proposed adjustments that could strengthen the work:
- There could be subsection describing different applications/benchmarks that could be derived from the datasets (in Fig 3), and possibly
provide access to these in interface to dataset
- Include more benchmark metrics in addition to total accuracy (could be included F1-score, example confusion matrix etc.)
- Introduce all abbreviations before usage; these are not clear for all readers (POV, IRB, BMI etc.)
- Current conclusion section is more like discussion section (or list of findings) than conclusion; it would be good have separate summary of the findings and final conclusions of the work

**Strengths And Weaknesses:**

Strengths
- Compared to previous works, there is a need for this kind of dataset
- Some benefits (as number of modalities, classes, number of subjects) compared to existing datasets
- Proposal includes all the necessary information access to data and running the chosen benchmarks
- Few applications/benchmarks and example use cases given with baseline comparison of DL approaches

Weaknesses
- A bit limited information about other use cases (referring to Fig. 3): different applications/benchmarks possibilities derived from the dataset could be described more detailed
- Details missing on data pre-processing; imputation, normalisation etc.
- Limited performance metrics in benchmark
- Current conclusion should be revised and divided on discussion and conclusions sections

---

### Review · Reviewer_Bk3D · 2025-06-01

**Recommendation:** 3
**Confidence:** 1

**Summary Of Contributions:**

This work introduces a new dataset, DARai, a multimodal and hierarchical dataset for human activity understanding. The proposed dataset is annotated at three levels of granularity: high-level, low-level, and fine-grained. Furthermore, the dataset considers both scripted and unscripted activities. The paper provides a detailed description of the dataset construction and evaluates the performance of various models on several tasks using the dataset.

**Strengths:**

- DARai is a large-scale dataset that introduces hierarchical categorizations of human activity video data including a wide range of activities
- The paper is generally written well and explain the significance of their contributions through both detailed dataset construction and benchmark evaluations.
- I believe this dataset would serve as a more robust and diverse benchmark in tasks such as activity recognition with more fine-grained information.

**Audience:**

Yes

**Claims And Evidence:**

Yes

**Datasets And Benchmarks:**

The dataset is described well including the annotations, number of samples, modalities, and structure. More information is also provided in the supplementary material.

**Extended Submissions:**

N/A

**Requested Changes:**

- Please be careful in the use of 'counterfactual' throughout the paper and make it clear what this exactly refers to since there is no causal perspective provided in this paper.

**Strengths And Weaknesses:**

## Strengths
- This paper introduces a refined multi-hierarchy dataset for human activity and has great utility for a wide range of computer vision tasks, including areas such as group activity recognition.
- The extensive empirical benchmarking study on the proposed dataset increases the value of this work.
- Incorporating both scripted and unscripted activities sets this dataset apart from the literature where only one or the other is considered.

## Weaknesses
- I'm not sure if 'counterfactual' is necessarily an accurate description of the overlap in activity hierarchies since its connotation is typically in causality.
- Although introducing hierarchies in datasets is important, I am not sure the L1, L2, L3 hierarchy is informative. For instance, what is the difference between L2 and L3? It could be seen as somewhat redundant information.

---

### Review · Reviewer_y85E · 2025-09-22

**Recommendation:** 3
**Confidence:** 2

**Summary Of Contributions:**

This paper introduces DARai (Daily Activity Recordings for artificial intelligence), a new multimodal and hierarchically annotated dataset for understanding daily human activities. The key contributions are:

- The dataset includes over 200 hours of recordings from 50 participants across 10 environments, incorporating data from 20 different sensors, including multiple cameras (RGB, Depth, IR), LiDAR, wearable IMUs, EMG, insole pressure sensors, biomonitors, and a gaze tracker. This represents a very sensor-rich dataset.

- Activities are annotated at three levels: high-level activities (L1), lower-level shared actions (L2), and fine-grained procedures (L3), with designed overlaps between levels to study compositionality.

- The data collection involves a hybrid of scripted high-level goals and unscripted execution, capturing natural variations. The design explicitly includes counterfactuals (e.g., moving light vs. heavy objects) to test sensor sensitivity.

- The authors provide extensive benchmark experiments on activity recognition, temporal localization, and action anticipation across all hierarchical levels, evaluating unimodal and multimodal fusion approaches.

- The experiments highlight the strengths and weaknesses of different modalities, including the impact of viewpoints, body side for wearables (cross-view/cross-body effects), and environmental differences.

- The dataset, annotations, and experimental code are made publicly available.

**Strengths:**

DARai helps address the need for richer, more complex datasets by providing a wide array of modalities and a hierarchical annotation scheme. The authors situate DARai effectively within the landscape of existing datasets (Tables 1 and 2), highlighting its novel aspects. The dataset should be of interest to researchers in human activity recognition, multimodal learning, robotics, and assistive technologies. The data collection and annotation appear carefully executed. The experiments are well-designed and provide a solid foundation for future work. The manuscript is clear and well-structured.

**Audience:**

Yes

**Claims And Evidence:**

Yes, claims are supported by convincing and clear evidence.

**Datasets And Benchmarks:**

Yes, the paper provides sufficient detail.

*Data Collection and Organization:* Section 3 and Appendix D.

*Availability and Maintenance:* Available on IEEE Dataport and GitHub. Long-term plan could be clearer.

*Ethical and Responsible Use:* Discussed and appears adequate.

*Reproducibility:* Supported by provided splits and code.

**Extended Submissions:**

This appears to be new work, not an extension of a previously published work.

**Limitations:**

*Sensor Setup Complexity and Cost:* The elaborate setup is expensive and complex.

*Hybrid Approach Constraint:* The setup necessitated a hybrid scripted/unscripted approach.

*Environment Coverage:* Limited to kitchen and living room environments.

*Participant Pool Size:* The pool of 50 participants is a moderate size.

**Requested Changes:**

- Further discussion on the class distributions within each hierarchy level (L1, L2, L3) and how potential imbalances are mitigated by the evaluation metric or might affect the interpretation of results would be valuable.
- Explicitly stating the long-term maintenance plan for the dataset on IEEE Dataport would be beneficial for the community.
- Exploring the impact of ablating sensor groups could provide more nuanced insights into modality contributions.
- Linking to a few video examples on the project website would help readers appreciate the data's richness.

**Strengths And Weaknesses:**

**Strengths:**

- The number and variety of synchronized sensors provide a valuable opportunity to study multimodal fusion.
- The three-level hierarchy with sharing of L2/L3 elements is promising for studying complex activity structures.
- The paper presents a good set of baseline results for various tasks, offering useful insights (summarized in Table 5).

**Weaknesses:**
- With 50 participants, the dataset might not fully capture inter-subject variability compared to very large-scale efforts, potentially affecting model generalization.
- The high cost (~$22,000) and complexity of the sensor setup are significant barriers to replication or extension by other research groups.
- Data collection is focused on kitchens and living spaces, somewhat limiting the diversity of daily activities and contexts.

---

### Review · Reviewer_AUA2 · 2025-10-15

**Recommendation:** 3
**Confidence:** 2

**Summary Of Contributions:**

This paper introduces DARai, a new large-scale hierarchical and multimodal dataset for human daily activity understanding. DARai includes 200+ hours of continuous recordings collected from 50 participants across 10 natural environments using 20 modalities from 12 sensors, including RGB/depth cameras, IMUs, EMG, insole pressure, biomonitoring, radar, and gaze tracking. Each recording is annotated at three levels of granularity to capture the hierarchical nature of human behavior. The dataset also introduces the notion of “action counterfactuals”, representing alternative executions of the same activity under different conditions.

**Strengths:**

See above.

**Audience:**

Yes

**Claims And Evidence:**

Yes.

**Datasets And Benchmarks:**

Yes.

**Extended Submissions:**

No.

**Limitations:**

See above.

**Requested Changes:**

See above.

**Strengths And Weaknesses:**

**Strengths:**
- DARai integrates a broad spectrum of sensors, offering one of the richest multimodal datasets for daily activity analysis to date.
- The authors evaluate cross-view, cross-body, unimodal, multimodal, and temporal anticipation tasks with consistent protocols and clear takeaways.
- The three-level structure (L1–L3) enables systematic study of granularity effects and temporal dependencies.

**Weaknesses and request changes:**

Generally, this is a good submission, but I still have some concerns.
- Lack the background of the participants, like age, gender and so on.
- The words and numbers in some figures are too small, especially in Figure 8, 11, and 24, making it unfriendly for readers of printed (paper) versions.
- Since transformer-based models have achieved great success in many areas, they are not included or discussed in the baselines of this submission.
- A typo in Appendix G.1.1 two lines above Table 19.